# A GPU-based caching strategy for multi-material linear elastic FEM on regular grids

**Christian Schlinkmann**[1,2], **Michael Roland**[2], **Christian Wolff**[1], **Patrick Trampert**[1,2], **Philipp Slusallek**[1,2], **Stefan Diebels**[2], **Tim Dahmen**[1]*

**1** Deutsches Forschungszentrum für Künstliche Intelligenz (DFKI) GmbH, Saarbrücken, Germany,
**2** Saarland University, Saarbrücken, Germany

* Tim.Dahmen@dfki.de

**Data Availability Statement:** Source code of the software is available at https://github.com/c3di/voxsol. All other relevant data are within the manuscript and its Supporting Information files.

## Abstract

In this study, we present a novel strategy to the method of finite elements (FEM) of linear elastic problems of very high resolution on graphic processing units (GPU). The approach exploits regularities in the system matrix that occur in regular hexahedral grids to achieve cache-friendly matrix-free FEM. The node-by-node method lies in the class of block-iterative Gauss-Seidel multigrid solvers. Our method significantly improves convergence times in cases where an ordered distribution of distinct materials is present in the dataset. The method was evaluated on three real world datasets: An aluminum-silicon (*AlSi*) alloy and a dual phase steel material sample, both captured by scanning electron tomography, and a clinical computed tomography (CT) scan of a tibia. The caching scheme leads to a speed-up factor of ×2-×4 compared to the same code without the caching scheme. Additionally, it facilitates the computation of high-resolution problems that cannot be computed otherwise due to memory consumption.

## Introduction

### Motivation

The method of finite elements (FEM) is a commonly used approach for solving partial differential equations, for example for the simulation of mechanical systems. The approach originates from engineering science, where objects are typically described as computer-aided design (CAD) models using higher order curves, for example trimmed non-uniform rational B-splines (NURBS). Classical FEM simulations require these higher order curves to be discretized in the form of a volume mesh that is composed of individual cells (the finite elements). Typical cell meshes have a hexahedral or tetrahedral topology. The mesh cells are flexible in their connectivity in order to express a wide range of shapes. The individual cells are mapped to standard elements of typically unit size, and mechanical laws are formulated through the use of basis functions. Most often, product spaces of polynomials of degree one to three (linear, quadratic, cubic) are used, but higher order degrees as well as more exotic functions such as NURBS have also been used successfully [1].

The simulated mechanical properties of each finite element are described by a local stiffness matrix, obtained by combining the element's specific material properties with the basis

**Funding:** T.D., C.S., P.S. The authors gratefully acknowledge the financial support of the German Bundesministerium für Bildung und Forschung (BMBF) under the grant 13GW0124. https://www.bmbf.de/ The funders had no role in study design, data collection and analysis, decision to publish, or preparation of the manuscript.

**Competing interests:** The authors have declared that no competing interests exist.

functions. The interconnectivity of the finite elements is then modeled by combining the local stiffness matrices into a single global stiffness matrix $A$. Typically it is symmetric, positive definite and often very sparse, as each finite element is typically incident to only a small number of other elements. A system of equations of the form $Ax = b$ models the complete mechanical system, where $b$ is the load vector containing volume forces as well as the boundary conditions and $x$ is a vector of unknown values, such as displacements. An overview of the notation is given in Table 1.

The resulting algebraic system may be solved directly, for example through Pardiso [2, 3] or Cholesky decomposition [4, 5]. However, for large systems this becomes infeasible and an iterative numerical approach is used instead, for example the conjugate gradient method. A best-guess starting point for the iterative solver may be approximated by first solving a coarser discretization of the same mechanical system. This may be repeated several times on successively coarser representations, creating a hierarchy of discretizations; a process known as the multi-grid method. In this publication we refer to these coarser discretizations as additional levels of detail.

As FEM simulations of problems of practically relevant size can be very computationally intensive, their parallelization has long been an active area of research. However, the requirements of parallelization schemes are often at odds with properties introduced to the finite element model by unstructured mesh geometries. For example, modeling the connectivity between elements in an arbitrary mesh introduces memory fragmentation, making efficient parallel access more difficult. To combat this effect, mesh elements may be reordered to maximize memory bandwidth [6, 7]. In multigrid methods, topology changes between two discretization levels in unstructured meshes can cause ambiguities that are difficult to handle [8–10].

More recently, FEM simulations are increasingly applied to models that do not originate from CAD software but are generated using three-dimensional imaging technologies such as computed tomography (CT) or three-dimensional microscopy. Applications for these scenarios are numerous. Biomechanical simulations are often based on clinical CT data [11, 12]. The mechanical simulation of microstructures in materials science and engineering can be based on three-dimensional electron microscopy techniques such as focused ion beam scanning electron microscopy (FIB/SEM) [13], electron-, or X-ray tomography [14].

These applications, together with continued advances in computing hardware, particularly in the field of general-purpose GPU computing, have renewed interest in the field of fixed grid finite element analysis. Here the problem is discretized by a regular grid of finite elements, typically hexahedra, having uniform size and orientation. This provides a basis for efficient and predictable memory access and eliminates the possibility of ambiguous topology changes in the multigrid method.

**Table 1. Symbols used in the description of the algorithm.**

| Symbol | Meaning | Dimension | Comment |
|---|---|:---:|:---:|
| $K$ | Local stiffness matrix at center vertex | 3×3 | $K = M_{13}$ |
| $M_i$ | Local stiffness matrix at neighbor vertex $i$ | 3×3 | $i \in (0..26), i \neq 13$ |
| $u$ | Local displacement at center vertex | 3×1 | |
| $d_i$ | Local displacement at neighbor vertex $i$ | 3×1 | $i \in (0..26), i \neq 13$ |
| $\varphi$ | External force vector at center vertex | 3×1 | Boundary condition |
| $\delta$ | Residual error at any vertex | Scalar | |
| $f$ | Local right hand side vector | 3×1 | |
| $\varepsilon$ | Convergence threshold | Scalar | |

The volume data generated in these applications typically has a high resolution. By constraining the shape and placement of finite elements to a fixed grid this resolution cannot be decreased without also coarsening the shapes that are being modeled, leading to inaccuracies in the simulation. When mapping the problem to GPU hardware this can result in very high memory consumption and memory bandwidth bottlenecks, as the global stiffness matrix becomes very large. To combat this problem, a matrix-free approach is often used [34], in which entries in the global stiffness matrix are computed on-demand. This trades an increase in computational effort for a reduction in memory bandwidth.

We contribute to the class of matrix-free FEM for linear elasticity problems by presenting a novel, GPU friendly caching scheme for linear elastic fixed-grid problems with a limited number of discrete materials. In this case, we observe that the global stiffness matrix has a specific, highly redundant structure. Many vertices in the problem share the same spatial configuration of materials and boundary conditions in the eight cells bordering that vertex. We call this the *local problem configuration*. The stiffness matrices for these local problem configurations contain constant terms, allowing them to be pre-computed and stored in L1 cache, thereby lowering overall memory bandwidth for multi-material problems.

## State-of-the-art

The most popular alternatives to FEM in elasticity are the finite difference methods (FDM). FDM are also a class of numerical methods for solving partial differential equations by approximating the derivatives with finite differences. FDM are widely used in numerical analysis and several engineering applications [15]. Even in the field of linear elasticity, FDM are still an active field of research, in particular with focus on staggered grids and locking-free concepts [16]. The extension to generalized FDM has also proven to be a good meshless method for elasticity problems [17].

Another alternative to the most common FEM simulations for linear elasticity problems is the virtual element method (VEM) [18]. Virtual elements can be almost analogous to FEM by using similar element spaces [19]. In contrast to classical FEM, the basis functions which model the material inside the finite elements are defined only implicitly. This enables the use of arbitrary polygonal elements within the same mesh, allowing more flexible meshing of complex geometries. It also requires no special handling of hanging nodes, as the two incident polygons sharing the hanging node can have different numbers of sides. However, the elemental stiffness matrix must be calculated with the help of consistency and stability terms to recover an approximation of the basis functions, since these exist only virtually [20].

Multigrid methods are algorithmic concepts using a hierarchy of discretizations in the solving process [21, 22]. To accelerate the convergence of the solver, an iterative global correction of the solution on the finest grid level is used in combination with the solution of a coarse problem [23]. Here, the coarsening process is recursively used until a grid level is reached where the problem can be solved very efficiently [24], which leads to a quicker distribution of information over the computational domain and a dampening of low frequency errors. However, this hierarchy of grid levels necessitates efficient transfer operators from one grid level to the next and back as well as representations of the discretized mechanical system on each grid level. In combination, the coarse grid correction, restriction of the residual error to coarser grid levels, projection of the computed correction to finer grid levels and smoothing of the errors results in a faster convergence rate [25–27].

Multigrid methods are also used with locally adaptive refined and coarsened grid levels. In this case, one distinguishes between level meshes, i.e. cells with the same refinement distance from the coarse grid, and leaf meshes consisting of all active cells of the hierarchy. These

concepts require specific algorithms and data structures [28]. In order to avoid the problems of multigrid methods with complex geometries, algebraic multigrid methods were developed acting only on the underlying linear system of equations and not on the grid itself [23].

As the performance bottleneck is often data movement rather than floating-point operations (flops), block-asynchronous multigrid smoothers on GPUs [29] perform more flops in order to reduce the synchronization effort. For many problems, asynchronous iteration strategies outperform the classical synchronized relaxation methods [30].

The algorithmic framework for matrix-free operator evaluation enables very efficient implementations because the coefficient matrix is not stored explicitly. Instead, the access to the matrix is realized by the evaluation of the underlying coefficients on the fly via matrix-vector products [31, 32]. Matrix-free FEM approaches have been proposed to reduce memory requirements and alleviate memory bandwidth bottlenecks in parallel computing [33]. Originally developed for distributed computing clusters, these approaches show renewed interest for general-purpose GPU applications. When working on a fixed grid with uniform element sizes and a single material, all finite elements share the same local stiffness matrix, which can be pre-computed [34]. Storing this matrix in constant memory further reduces bandwidth requirements in the matrix-free GPU approach [35].

Per-vertex approaches, also known as node-by-node approaches, are a re-formulation of the finite element problem with a focus on solving for individual nodes, rather than finite elements [36]. The closest related work to our approach is based on a per-vertex Gauss-Seidel relaxation [37]. By shifting the computational focus away from elements and onto per-vertex equations, vertices are iteratively updated by considering the updated displacements of their direct neighbors. Our work improves over [37] for the class of multi-material problems by introducing a highly efficient data representation that exploits self-similarity in the system matrix, hereby reducing memory consumption and enhancing cache efficiency by several orders of magnitude.

## Material and methods

### Problem

Our method solves the following standard linear elasticity problem. Assume that the boundary of the elastic body is divided into two disjoint sets $\Gamma_D$ (Dirichlet boundary) and $\Gamma_N$ (Neumann boundary) and assume that a system of body forces $\boldsymbol{f} : \Omega \to \mathbb{R}^3$ and surface tractions $\boldsymbol{g}_N : \Gamma_N \to \mathbb{R}^3$ act on the body. On the other part $\Gamma_D$ of the boundary, the body is partially fixed in space. Under the assumption of small displacements, the displacement $\boldsymbol{u} = (u_i)_{1 \leq i \leq 3}$ satisfies the following problem:

$$-\sum_{j=1}^{3} \frac{\partial \sigma_{ij}}{\partial x_j}(\boldsymbol{u}) = f_i \text{ in } \Omega,$$

$$u_i = 0 \text{ on } \Gamma_D,$$

$$\sum_{j=1}^{3} \sigma_{ij}(\boldsymbol{u})n_j = g_i \text{ on } \Gamma_N,$$

where $\boldsymbol{n} = (n_i)_{1 \leq i \leq 3}$ is the unit outward normal to the boundary $\Gamma_D$, $f_i$ and $g_i$ are the components of the forces $\boldsymbol{f}$ and $\boldsymbol{g}_N$, and $\sigma_{ij}(\boldsymbol{u})$ is the stress tensor, cf. [38]. The problem is then discretized in the standard way in finite element analysis. Therefore, standard notations for

Lebesgue and Sobolev spaces are used. Let $(\cdot,\cdot)$ denote the inner product in $(L^2(\Omega))^d$, $d \geq 1$. Find $\mathbf{u} \in H^1(\Omega)^3$ such that

$$a(\mathbf{u}, \mathbf{v}) = \langle F, \mathbf{v}\rangle \text{ for all } \mathbf{v} \in H_D^1(\Omega)^3, \text{ with the scalar product}$$

$$a(\mathbf{u}, \mathbf{v}) = \int_\Omega \sigma(\mathbf{u}) : \boldsymbol{\varepsilon}(\mathbf{v})d\mathbf{x} + \int_{\Gamma_N} \mathbf{g}_N \cdot \mathbf{v} \, ds \text{ and the bilinear form}$$

$$\langle F, \mathbf{v}\rangle = \int_\Omega f \cdot \mathbf{v} \, d\mathbf{x}.$$

## Algorithm

Our approach (Fig 1) is based on a number of assumptions about the mechanical problem. In most problems with a limited number of discrete materials [11, 38, 40], the global stiffness matrix contains many redundant entries. Firstly, there are often large regions of uniform material. In the case of a fixed-grid with uniform finite element size and shape, this corresponds to blocks in the FEM matrix that are identical up to relatively simple index changes. Secondly, even if the materials around any vertex are not uniform, the same configuration of boundary condition and local material distribution should appear many times in the problem, also leading to a redundant matrix structure. Experimental support of this assumption is shown in the results section.

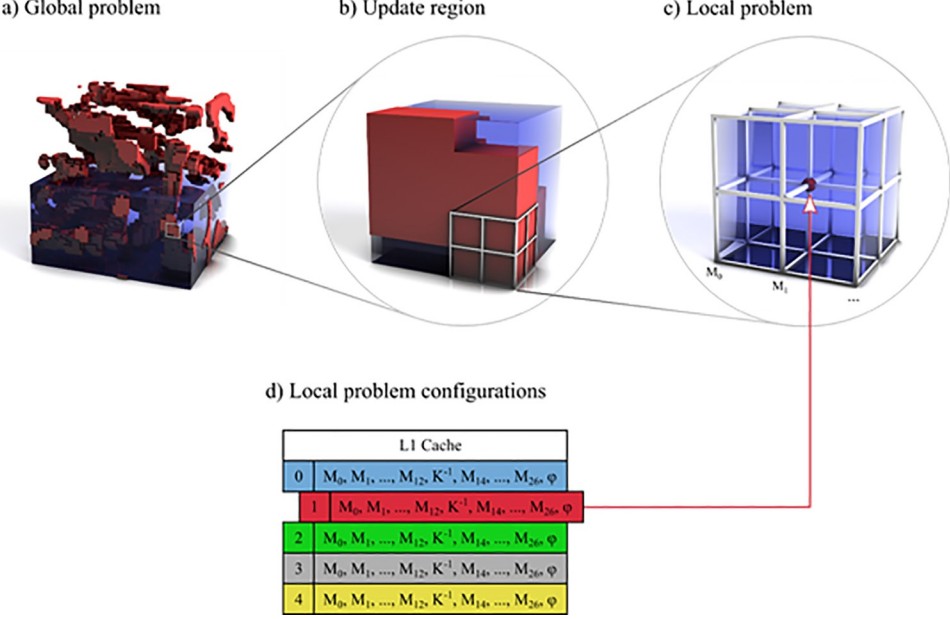

**Fig 1. Overview of our approach.** a) Two distinct materials are represented with red and blue. The solution is updated by locally working on update regions. The number and size of the update regions depends on the GPU architecture. The problem is solved by selecting and processing update regions in a loop until a convergence criterion is reached. b) Vertices in the update regions are transferred to shared memory. The update regions are processed by repeatedly updating the position of a number of randomly selected vertices. When the update region is locally converged, the result is transferred back to global memory. c) For the purpose of updating a vertex position, we define a local problem as follows: A local problem corresponds to solving for the position (x,y,z) of a center vertex, while assuming the positions of all 26 neighboring vertices are fixed for the duration of this update step. d) We solve for the center vertex position by assigning a configuration ID depending on its boundary conditions and the layout of materials of the eight incident voxels. We call this the local problem configuration, with each unique configuration above represented by a different color. For each configuration, local stiffness matrices for the 27 vertices in the local problem are pre-computed and stored in a cache friendly format. The key observation is that for large problems with an ordered distribution of discrete materials, many vertices will share the same configuration.

In a vertex-based approach, each vertex can be regarded as the center of a 27-vertex local problem (Fig 1C). In this view, the 26 outer vertices can be treated as fixed while the center vertex is free to move to a new equilibrium point in relation to its neighbors. Solving this local system requires the stiffness matrices and displacement vectors of all 27 vertices, as well as any boundary conditions assigned to the center vertex. If the center vertex is part of the problem boundary, then some neighboring vertices may not exist. In this case, the missing vertices are substituted with a null stiffness matrix. Crucially, in the case of linear elasticity, the matrices and boundary conditions for this vertex remain constant for all subsequent relaxations.

$$u = K^{-1} \cdot \left( \left( \sum_{\substack{i = 0, \\ i \neq 13}}^{26} M_i d_i \right) + \varphi \right) \tag{1}$$

Eq 1: The stiffness matrices M of the 26 outer vertices are multiplied with their displacement vectors $d$ and summed to form the right-hand side (RHS) vector. Index 13 is understood to be the center vertex, which should not contribute to the RHS as it is the vertex being solved for. If applicable, external incident forces $\varphi$ are added. To solve the system, the pre-inverted stiffness matrix $K^{-1}$ of the center vertex is multiplied with the RHS vector, giving the new displacement vector $u$ of the center vertex. The pre-calculation of $K^{-1}$ is numerically stable, as K is very small (3×3) compared to the global system matrix.

The algorithm begins by visiting each vertex of the global problem to examine the distribution of materials in the eight incident voxels, as well as the boundary conditions of the center vertex. Together, this information is called the *local problem configuration*. If the configuration is known, the vertex is assigned the respective ID, otherwise a new ID is generated and assigned. Assuming the number of configurations is below 65,536, each vertex requires a 2-byte configuration ID and a three-component displacement vector with either single or double precision.

Once all vertices have been visited, the 27 3×3 stiffness matrices for each unique local problem configuration are computed from the material properties of the eight voxels. Each unique local problem configuration also contains a three-component vector describing the external force incident at the center vertex, corresponding to any Neumann boundaries that were assigned to this vertex. Vertices that are exclusively surrounded by voxels containing no material, subsequently referred to as *void voxels*, are assigned an invalid ID and are later ignored during the update phase. Vertices that are assigned Dirichlet boundaries are also assigned an invalid ID. This ensures that the vertex retains its initial displacement throughout the simulation.

In order to map the algorithm to GPU hardware, a large number of vertices are processed in parallel and results are exchanged asynchronously. However, in order to make the most efficient use of memory bandwidth and caches, we must ensure data locality at the hardware level. This means clustering vertices as tightly as possible, however in a parallel setting this can lead to race conditions. This dilemma is solved using the following multi-color partitioning scheme also used in [37], which ensures that no two neighboring vertices are updated in parallel, while still allowing parallel updates to many vertices inside each block.

The partitioning scheme places vertices into eight distinct subsets, with the $k$-th vertex of each voxel being placed into the $k^{\text{th}}$ subset. This process is repeated for all vertices in the update region. This pattern ensures that no two vertices in the same subset are adjacent to one another. The eight partitions are updated sequentially while the vertices inside each partition may be updated in parallel. This partitioning refers to the order of computation and should not be confused with element connectivity.

To further improve convergence speed, a multigrid method is employed. For each level of detail, the fixed-grid approach allows eight voxels to be combined into one. This results in an eightfold reduction in problem size with each successive level of detail. The fixed-grid approach also allows materials, displacements, and boundary conditions to be mapped directly from one level to another. Coarse grid voxels are assigned a material by majority vote of their eight children. Ties are broken by choosing the material with the larger ID. Additionally, void materials are not considered in the majority voting, meaning a coarse grid voxel is only assigned the void material if *all* of its children are also void. At each level of detail, the multi-grid structure is then used in the conventional way: an approximate starting point for the iterative solution is found by first solving a coarser level and projecting the resulting displacements to the finer level.

## Implementation

The algorithm was implemented using C++ and Nvidia CUDA 10. A fixed-grid, hexahedral representation of the mechanical problem is generated and stored on the host. This representation is called the *discrete problem*, which holds the material assignments of all hexahedral elements as well as the boundary conditions of the problem. The discrete problem is an intermediate representation not used in the iterative solver, instead serving to assist in the pre-processing of the multigrid structure and the local problem configurations. A vertex-based representation is generated from the nodes of the discrete problem in which vertices, rather than hexahedral elements, are stored. Local problem configurations are referenced by the configuration IDs stored in the vertices. Each configuration consists of 27 matrices of dimension 3×3 and one three-component vector for the Neumann stress incident at the faces connected to the center vertex of the local problem. The total number of configurations is heavily dependent on both the number and distribution of discrete materials in the problem. The configurations are pre-computed on the host and stored in texture memory on the GPU, after which they are treated as read-only. These steps are repeated for each level of the multigrid, down to a minimum problem size encompassing the size of one update region. We employ a half V-Cycle multigrid as the simplest possible approach that is compatible with our solver. Each level of the multigrid is solved completely before projecting the resulting displacements down to the next level. Once a coarse level has converged and the result projected to the next finer level of detail, the associated data may be discarded. In principle, more elaborate multigrid cycle schemes such as F-Cycles or W-Cycles are also compatible with our solver.

For each level of detail, the algorithm consists of two nested iterative loops, one on the host and one on the GPU. The host loop is responsible for selecting update regions and managing CUDA kernel launches. The number of update regions per host iteration is selected based on problem size, hardware capabilities, and the strategy used to distribute the update regions inside the problem. For simplicity, we will assume update regions are placed sequentially and non-overlapping, such that all vertices in the problem are covered. An overview of the simulation loop is presented as pseudocode (Fig 2).

On the GPU, each thread block first copies the vertices of its update region into shared memory (Fig 3A). It is important to note that an additional 1-vertex border on all sides must also be included, as the vertices at the edges of the update region require the displacements of neighbors that lie outside the update region. The vertex data is updated frequently and must be available to all threads in the block during construction of the RHS vectors, so these are stored in shared memory. The local problem configurations, on the other hand, are read-only and should be cached as efficiently as possible in the L1 cache of each streaming multiprocessor.

Each thread block is divided into groups of nine warps with 32 threads each, henceforth called a *worker* (Fig 3B). The number of workers is chosen to reflect the capabilities of the

```
while ( not converged() )
        regions = select_update_regions( number_of_update_regions );
        cuda_launch( region:regions as threadblock ):
                __shared__ local_vertices = copy_vertices_to_shared_memory( region );
                ( number_of_workers, iterations ) := select_hardware_specific_optimum();
                parallel for number_of_workers:
                sequential for iterations:
                        vertex = select_vertex( local_vertices );
                        matrices = fetch_configuration_from_cache( vertex );
                        rhs = assemble_right_hand_side( vertex, matrices );
                        lhs = matrices[center_vertex_index];
                        __syncthreads( region );
                        local_vertices[vertex] = solve_3x3( rhs, lhs );
                        copy_vertices_to_global_memory( region, local_vertices );
```

**Fig 2. Pseudocode of the central update loop.**

hardware with more workers allowing more vertices to be updated in parallel inside the update region. For simplicity, we will assume nine workers for the remainder of this paper.

Each worker is comprised of a single warp containing 32 threads. To ensure coherence, we disable 6 threads in each warp using the *__ballot_sync* intrinsic, leaving 26 threads to assemble the RHS entries for each of the 26 neighbors in the local problem. The update is performed by the following computations: (1) Each worker is assigned a vertex in the active partition of the update region. (2) For each of the 26 neighboring vertices, one thread is used to compute the RHS contribution by multiplying the current displacement with the neighbor's corresponding stiffness matrix stored in the local problem configuration. (3) The 26 contributions are summed using the CUDA *__shfl_down_sync* intrinsic, which sums the values of a warp in-register. (4) The first three threads of each warp then place the summed contribution into the appropriate RHS vector component in shared memory. (5) A block-wide *__syncthreads* call ensures that the RHS components of all workers are complete before moving on. This is required to avoid race conditions when transferring the new displacements back into shared memory. (6) Three threads are selected to calculate the *x*, *y*, and *z* components of the new displacement by adding the Neumann stress to the RHS and multiplying with the pre-inverted

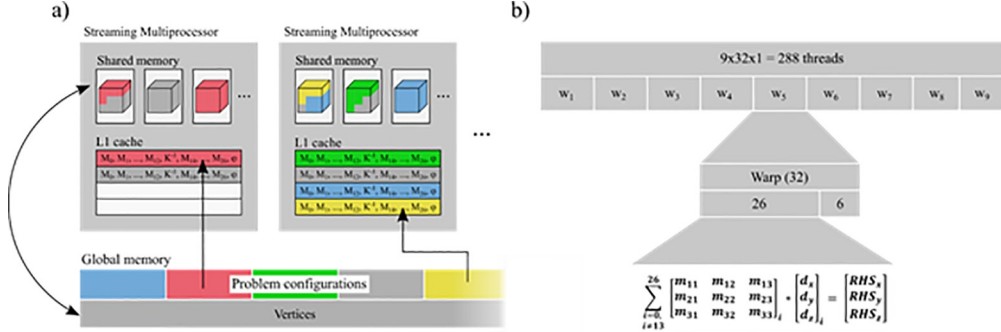

**Fig 3. The software architecture is driven by GPU hardware capabilities.** a) Memory layout. Vertex displacements of active update regions are stored in shared memory of each thread block. Problem configurations are stored in global memory. As they remain constant, they can be cached in L1. b). Thread layout. A thread block corresponds to an active update region. The thread block is subdivided into nine warps called "workers". Each worker updates one vertex at a time by assembling the right-hand side (RHS) and multiplying it with the inverted stiffness matrix of the center vertex. The RHS is assembled in parallel using 26 threads in each warp, one for each neighboring vertex.

center stiffness matrix, which is also stored in the local problem configuration. The new displacement vector is transferred to shared memory, available for the next iteration.

We assign nine workers per thread block, giving a total of nine warps or 288 threads per block. We chose to use update regions of 6×6×6 vertices, which expands to 8×8×8 vertices with the required 1-vertex border. The multi-color partitioning results in eight distinct partitions of 27 vertices each. The border vertices are not considered for partitioning since they are never updated. Thus, nine workers are able to update each partition in three steps, updating nine vertices in parallel per step. The vertices in each region are updated a total of three times to help ensure the entire update region has converged to its new equilibrium. The thread block then terminates after the new displacements are transferred from shared memory back to global memory.

## Determining convergence

Determining a suitable termination criterion is an important aspect of numerical approaches to solving systems of equations. Such criteria are often defined in terms of minimizing an error function to a specified lower threshold value $\varepsilon$, after which the solution is deemed to have converged. For the solver presented in this paper we define the error function as the relative error of the equilibrium condition $Ku = f$:

$$\delta = \frac{\|Ku - f\|_2}{\|f\|_2}, \qquad \|f\|_2 \neq 0 \tag{2}$$

We will refer to this term as the *residual error* of a vertex. In cases where $\|f\|_2 = 0$ the residual error is set to zero. Residual errors are sampled by scheduling a periodic update of the vertices in the solution. As this process does not write results back into the solution there are no data dependencies between vertices, so all vertices inside an update region may be processed in parallel. Furthermore, to reduce both the computational effort and the storage requirements by a factor of eight the residual errors are only calculated for every second vertex in each spatial dimension.

Each vertex is assigned one thread to complete the residual error calculation as defined in (2). The result is stored in a separate array, of which two copies exist. One holds the result of the previous residual error pass $\delta_{k-1}$ while the other holds the result of the current pass $\delta_k$. These arrays are alternated between kernel executions.

$$\delta_{total} = \frac{1}{N} \sum_{\substack{j \in (0..N), \\ \delta_j \neq 0}}^{N} \delta_j^{k-1} - \delta_j^k$$

The final convergence criteria $\delta_{total} < \varepsilon$ for the solution is used to determine if the iterative procedure can be terminated. However, when taking the mean of the residual errors in this way, two things must be considered. As void voxels have a fixed displacement of zero and are ignored during the update phase, they should not contribute to the overall error of the solution. Furthermore, in the initial iterations of the solution the only vertices with a non-zero residual error are those directly affected by the external forces. Taking the mean of all vertices can therefore terminate the solution prematurely. For these reasons, all vertices with a residual error of exactly zero are ignored when calculating the residual error of the solution.

## Datasets used for evaluation

Three datasets were used for evaluation purposes (Fig 4). The first consisted of a materials science sample of aluminum with embedded silicon (*AlSi*) coral-like structures, imaged using

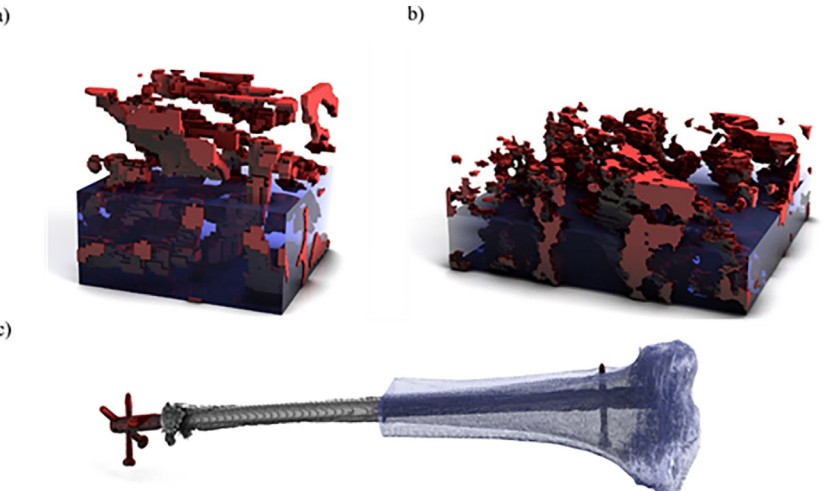

**Fig 4. Datasets used for the evaluation.** a) A material probe obtained through FIB/SEM containing silicon pockets embedded in aluminum. b) A sample of dual-phase steel captured through serial sectioning. c) A tibia bone with a distal fracture and intermedullary nail captured by computed-tomography (CT) scan.

focused ion beam scanning electron microscopy (FIB/SEM) [41]. The dataset contains subset of 64×64×64 voxels, with a voxel size of 46×46×46 *nm*, for a total of 262,144 voxels. Material properties of the two phases were assigned from literature values [39]. For the aluminum phase, a Young's modulus of 70 GPa and a Poisson ratio of 0.34 are assumed, for the eutectic silicon a Young's modulus of 107 GPa and a Poisson ratio of 0.27. The bottom of the sample was fixed in *z* direction while two sides were fixed in *x* and *y* directions respectively. A compressive force of 6 Newtons was uniformly distributed over the top face of the sample.

The second dataset (*dual phase steel*) consisted of a larger materials science sample of two-phase steel consisting of a ferrite matrix with martensite inclusions, captured through serial sectioning scanning electron microscopy with a voxel size of 411×411×407 *nm*. The resolution of this dataset was 256×256×100, for a total of 6,553,600 voxels. The ratio of ferrite to martensite was approximately 90:10 respectively. A Young's modulus of 200 GPa for the ferrite phase and 220 GPa for the martensite phase were used. Both phases were assigned a Poisson ratio of 0.27. The sample was fixed analogous to the *AlSi* dataset and a compressive force of 1000 Newtons was uniformly distributed over the top face of the sample.

The final dataset (*tibia*) was a computed tomography (CT) scan of a human tibia with a complex fracture and an intermedullary nail. The implant was assigned material properties consistent with titanium, with a Young's modulus of 108 GPa and a Poisson ratio of 0.342. The tibia bone was manually segmented into two distinct materials, cortical and cancellous bone, based on the grey values found in the CT data. Additionally, both the fracture gap and callus were segmented as separate material classes to allow the evaluation of strains in the fracture area. Including the void material this gives a total of 6 distinct material classes. Cortical bone was assigned a Young's modulus of 18 GPa and a Poisson ratio of 0.3, cancellous bone 590 MPa and 0.3 respectively. The fracture gap and callus were assigned a Young's modulus of 5 MPa and a Poisson ratio of 0.4. After preprocessing and cropping, the dimensions of the *tibia* dataset were 169×281×1334, for a total of 63,350,326 voxels, with a voxel size of 0.44×0.44×0.3 mm. A compressive force of 902 Newtons was applied to the tibial plateau to simulate a standing body weight of 92 kg. The distal end of the tibia was fixed in all directions while the tibial

plateau was fixed in $x$ and $y$ directions. Note that a large number of voxels (approximately 56 million) surrounding the bone are filled with void material.

## Evaluation environment

Evaluation was performed on a headless GPU server with an Intel Xeon E5-2680, 256 GB of RAM and one Nvidia Tesla P100 with 16 GB of memory. The P100 includes 56 streaming multiprocessors (SM) each with 24 kB of L1 cache.

As a baseline, each dataset was also simulated using the iterative solver in the commercially available software Abaqus. Datasets were not re-meshed for this purpose. The simulation was performed in parallel on a workstation with 16 CPU cores and 128 GB of RAM. The linear solver was configured to use the iterative method with convergence criterion of $5.0 \times 10^{-3}$ for the average flux norm and $1.0 \times 10^{-2}$ for displacement corrections.

## Results

### Performance

The performance of the proposed caching strategy was evaluated on the three datasets described above. The experiments were performed using single precision with a convergence criterion of $1.0 \times 10^{-6}$

When the proposed caching strategy is disabled, each vertex is assigned a unique local configuration ID, which results in the required stiffness matrices being pre-computed and stored for all vertices in the problem. The algorithm then corresponds to the one proposed in [37]. In all cases the time taken to read the dataset was not included.

With two additional levels of detail the *AlSi* dataset took 13.7 s with caching disabled and 6.8 s with the proposed caching scheme enabled (Table 2). This represents a speedup of factor two. The iterative solver in Abaqus® was able to compute a solution in 17 s. An analysis of the corresponding simulation precisions is presented below.

With three additional levels of detail the *dual phase steel* dataset took 63.7 s with caching disabled and 14.3 s with the proposed caching scheme enabled; a speedup of factor 4.5. The iterative solver in Abaqus® required 17 minutes for a solution.

With five additional levels of detail the *tibia* dataset could not be completed with caching disabled. After 51 minutes of computation time the available GPU memory was exhausted upon transition to the full resolution detail level. A comparison with Abaqus® also failed due to insufficient system memory. Using the proposed caching scheme, the full dataset was completed in 8.9 minutes.

### Convergence properties

To verify the correctness of the converged solutions, the *AlSi* dataset was compared to a ground-truth solution generated by Abaqus®. Various voxel sizes were experimented on,

**Table 2. Comparison of execution times of our solver with and without the proposed caching scheme enabled.**

| Dataset | Number of elements | Abaqus | Caching disabled | Caching enabled | Speedup factor |
|---|---|---|---|---|---|
| AlSi | 262,144 | 17 s | 18.6 s | 9.7 s | 2x |
| Dual Phase | 6,553,600 | 13 min | 77.2 s | 19.6 s | 4x |
| Tibia | 65,350,326* | n/a | n/a | 8.9 min | n/a |

The tibia dataset could not be computed with caching disabled due to memory exhaustion. Results from the commercial software Abaqus were generated with the iterative solver using 16 CPU cores. *A large number of elements (~56 million) are filled with void material and not actively processed during the solution phase.

including anisotropic voxel sizes in each of the three degrees of freedom. In addition, both the magnitude and direction of the applied force was varied. In all cases the mean squared error over all vertex displacements was below 4%, using $1 \times 10^{-7}$ as the convergence criterion. To facilitate accurate comparisons with the Abaqus® solution, these experiments were performed using double precision using a higher convergence criterion and no additional levels of detail. Thus, the total computation time on the *AlSi* dataset is significantly longer than the results previously presented.

As our solver showed a discrepancy to the Abaqus solution of at least 3% the following experiment verified correct convergence to an analytically known problem. The test problem consists of a steel beam with dimensions 10m×1m×1m, elastic modulus of $210 \times 10^9$ and Poisson ratio of 0.3. A compressive load of $-1.0 \times 10^9$ N was simulated parallel to the longest axis. In this standard example the maximum displacement of the beam is:

$$d_{max} = \frac{F * L}{E * A} = \frac{-1 \cdot 10^9 * 10}{210 \cdot 10^9 * 1} = -0.047419m$$

A discretization of 100×10×10 elements was used with no additional levels of detail under a variety of convergence criterion (Fig 5C). Maximum displacement differed by 0.88% with a convergence criterion of $1.0 \times 10^{-6}$, falling to 0.15% at $1.0 \times 10^{-9}$ using double precision.

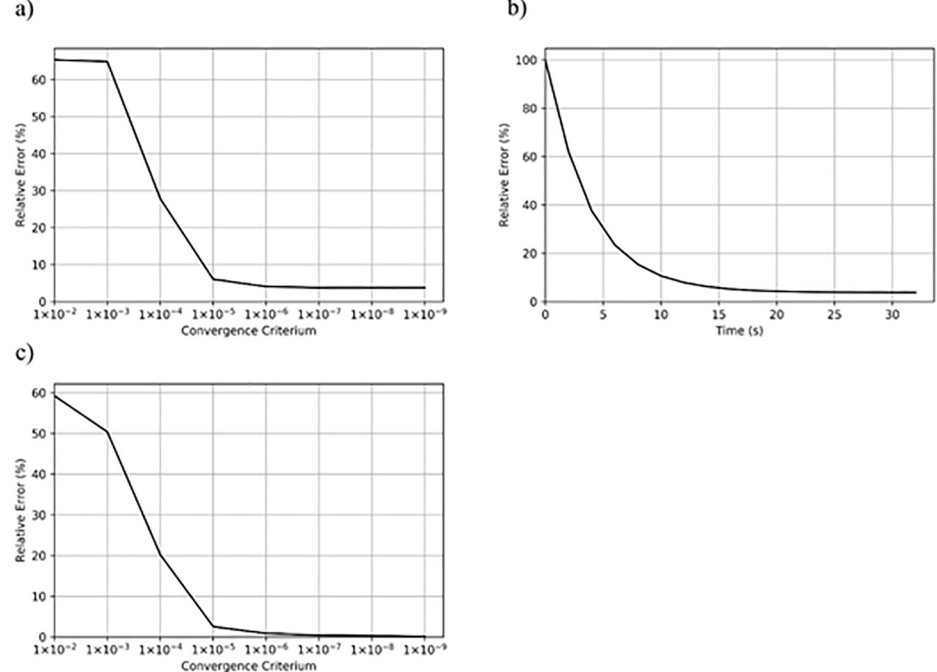

**Fig 5. Effect of convergence criterion on overall solution accuracy.** a) A solution for the AlSi dataset was computed with Abaqus and considered as ground-truth for this purpose. The mean relative error of the displacements as compared to this ground-truth solution was calculated on the full resolution detail level and plotted as a function of the convergence criterion. At 6% error a value of $1.0 \times 10^{-5}$ already gives a good approximation of the final solution, with the overall error stabilizing at 3.7% for a value of $1.0 \times 10^{-7}$. b) Mean relative error of the displacements as a function of simulation time, using no additional levels of detail and $1.0 \times 10^{-7}$ as convergence criterion. c) Relative error of the maximum displacement in an analytical example consisting of a steel beam with a compressive load applied parallel to the longest axis. The error continues to decrease as the convergence criterion is lowered, with a value of 0.46% using $1.0 \times 10^{-7}$ and 0.15% using $1.0 \times 10^{-9}$. These experiments were performed with double precision.

The convergence criterion is directly related to both the final accuracy and total computation time of the simulation. In cases where a fast approximation of the solution is sufficient, a large convergence criterion can be chosen to dramatically reduce computation time.

## Distribution of configurations

One of the assumptions underlying the presented approach is that in cases with a limited number of discrete materials, a small number of unique local problem configurations should suffice to describe the majority of vertices in the problem. Indeed, the three datasets examined in this work do exhibit this property (Fig 6). In the case of the dual phase dataset just two configurations are sufficient to describe 90% of the vertices in the problem. However, we also see that in the *tibia* dataset significantly more configurations are required to reach the same percentile. Here, the six unique materials and the irregular distribution of cortical and cancellous bone throughout the dataset result in a higher number of configurations overall. Still, in this challenging case just 41 out of a total 5897 configurations are sufficient to describe 90% of the vertices in the problem, even after filtering out all vertices which are surrounded by void materials.

## Impact of increasing number of materials

The impact of the proposed caching scheme on a problem depends on the number and spatial distribution of materials. We used three real life examples to evaluate this effect in practice. To get a better understanding on the robustness of the caching behavior under different, less favorable conditions, we also modified the *dual phase steel* dataset by artificially introducing

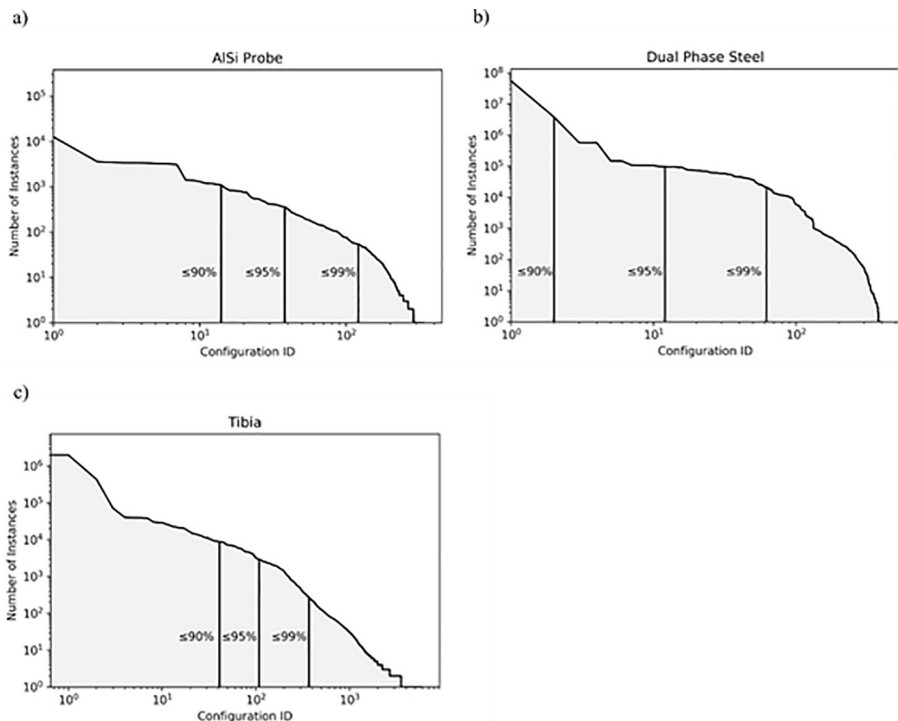

**Fig 6. Distribution of local problem configurations in the three datasets.** For each dataset, the 90, 95 and 99 percentiles are shown. Both axes have been logarithmically scaled. For both material probe samples, a small number of unique configurations are sufficient to describe 90% of the vertices in the problem. In the tibia case the combination of four distinct materials in a highly irregular fashion leads to a higher number of unique configurations. In all cases, voxels corresponding to void materials were filtered out.

additional materials. We distributed these additional materials along the *x* axis of the problem, such that slices of unique materials are created in the *y-z* plane. As a worst case we also investigate assigning each voxel a random material.

The additional materials all have the same material parameters as the ferrite steel phase in the original dataset but are treated as unique materials throughout the software. This is done to isolate the effect of decreased cache efficiency from any influence that varying material properties might also have. For example, softer materials lead to higher deformations which would also affect total simulation time. In addition, tests of materials with very large differences in elasticity ratios, for example on the order of $10^7$, have shown similar computational errors as in other FEM approaches. For such cases a deflation preconditioner may need to be used. The boundary conditions and additional levels of detail are analogous to the original dataset.

The efficiency of the L1 cache decreases slightly from 99% to 98% as the number of materials increases (Table 3). The effect on total run time is at the border of measurement precision with a slight increase from 12 s to 13 s in the 256-material case. In the case of 256 materials distributed randomly throughout the ferrite phase L1 cache efficiency drops considerably to 71% and the total run time of the simulation approaches the upper bound where the caching scheme is disabled entirely.

## Numerical precision

In order to investigate questions of numerical precision, we repeated the experiments on all three data sets (AlSi, Dual Phase, Tibia) in both single and double precision. For the used convergence criterion, we found no measurable difference in the precision of the result. Performance was observed to be less than 10% slower when using double precision.

## Technical performance

For the purposes of discussing technical performance we focus on the *dual phase steel* dataset with no additional levels of detail. The Nvidia Visual Profiler was used to investigate various performance metrics on the GPU during execution. A total of 448 update regions were processed per kernel execution, corresponding to eight thread blocks per available SM. Each thread block consists of nine warps of 32 threads each.

The main compute kernel, responsible for updating vertices in each update region as described in the implementation section, was found to exhibit 56% occupancy with four active blocks per SM, limited by total register count. With a 99.49% L1 cache hit rate for global memory accesses the scheduler in each SM averaged 3.7 active eligible warps per cycle. Average SM

**Table 3. Impact of increasing numbers of materials on cache efficiency and execution time.**

| Case | Number of configurations | L1 cache efficiency | Total runtime |
|---|---|---|---|
| Dual phase steel (unmodified) | 376 | 99% | 19 s |
| Dual phase steel (64 materials) | 1,161 | 99% | 22 s |
| Dual phase steel (256 materials) | 2,313 | 98% | 22 s |
| Dual phase steel (256 materials, random distribution) | 6,670,944 | 71% | 75 s |
| Caching disabled | 6,670,949 | 71% | 77 s |

All experiments were done with 4 additional levels of detail. Shown are the number of unique local problem configurations and observed L1 cache efficiency on the full detail level, as reported by the Nvidia Visual Profiler. When materials are distributed in an orderly fashion (here along the x-axis of the problem) the resulting impact on the caching strategy is minimal. With 256 materials distributed randomly throughout the problem the result closely matches the case with the proposed caching strategy disabled.

efficiency was 90.5%. Stalls were attributed primarily to memory dependency (22%), execution dependency (17%) and synchronization (7%).

## Discussion

The closest related work to our study is presented by Dick et al. in [37]. However, there are several key differences in the two approaches. The previous work stores all data required to perform the FEM simulation explicitly for both elements and vertices. This includes the elastic modulus, density and rotation matrices for elements, and for vertices an external force vector, Dirichlet conditions, the 27 3×3 local stiffness matrix coefficients, displacement vector, residual vector and RHS vector. This results in a considerable memory requirement of 1 KB per vertex using single precision, which the previous work attempted to mitigate through explicit indexing of finite elements to avoid allocating space for void regions in the problem.

In contrast, our approach requires no data to be stored on the GPU for the finite elements themselves and minimizes the memory requirements for vertex data through the proposed caching scheme. Using single precision, the 63 million vertices in the *tibia* dataset require just 1,013 MB of global memory on the GPU, with an additional 5,793 kB to store the 5,897 unique local problem configurations. The residual errors, stored one level of detail coarser than the corresponding problem, require just 128 MB of global memory. Together this represents an amortized memory requirement of just 18 bytes per vertex. For a single GPU with 8 GB of memory this gives a theoretical maximum size of approximately 400 million voxels for problems typical of the examples presented in this paper.

By assigning an invalid local problem configuration ID to vertices that are surrounded entirely by void materials these can be ignored during the update phase. Together with the reduction in memory requirements, this allows the use of implicit indexing based on the structured regular grid, which both simplifies the algorithm and allows for more predictable memory access patterns on the GPU.

Furthermore, by leveraging both the L1 cache and shared memory each vertex in an update region can be processed several times before the resulting displacements are written back into global memory. This allows the entire update region to converge to the local equilibrium over several steps, each time considering the deformation of the previous step, without requiring additional global memory bandwidth. We find that up to two additional updates to each vertex are beneficial, after which the active vertices in the update region are typically in equilibrium with the fixed vertices of the update region's border.

Experiments on using different numeric precisions (single vs. double precision floating point) showed no impact on the simulation results. This shows that for our specific experiments, the simulation precision is limited by the used termination criterion, not numeric precision. It also hints at a favourable numeric stability of our method, i.e. a relatively low numerical precision is required for a system matrix of given condition number.

Computation times were about 10% slower when using double precision. This slowdown originates entirely from the increased memory consumption, as the used P100 GPU is designed for double precision arithmetic and computation operations do not benefit from reducing floating point numbers to single precision. We speculate that the results would look differently on consumer hardware and that using double precision would have a much larger performance impact on these systems.

In order to give a baseline for the performance of our approach, we compared the implementation to the commercially available Abaqus® software. We found that the performance of our approach is superior in the defined case. However, one has to keep in mind that our

approach is limited to linear elastic systems on regular hexahedral grids while Abaqus® offers a much wider set of features.

While the experiments on the increasing numbers of materials are somewhat artificial, they serve to show that the caching strategy continues to be beneficial in cases where the number of materials increases as long as the assumption of an orderly distribution of materials still holds. However, a random distribution of a high number of unique materials defeats the caching strategy and nullifies any performance benefits. However, this case is quite rare in the fields of material science and medical CT images which comprise the primary focus of this work.

The convergence time of the presented solver is heavily dependent on the type of problem being solved. Elongated problems with forces acting parallel to the longest edge, such as the *tibia* dataset, tend to take longer to converge with respect to the total number of voxels. In addition, we found that external forces resulting in a pronounced lateral bending of long objects result in higher simulation times. This is not surprising as the successive update strategy favors shorter distances, for example in the *dual phase steel* dataset. Future work may help to mitigate this effect, for example by determining a better starting estimate for the solution or by improving the accuracy of the multigrid method for extremely coarse levels of detail.

A last consideration is that the caching scheme currently assumes that the material properties of the object in question remain constant throughout the simulation. However, in many physical systems this assumption does not hold, for example in the case of plastic deformations.

## Conclusions

We have presented a new caching strategy for finite element analysis on GPU for systems generated from computer-aided imaging techniques such as CT scans and scanning electron microscopy. We show that the regular grid structure of such datasets can be exploited to dramatically increase both memory and cache efficiency on GPU. This allows for the simulation of very large datasets on consumer grade workstations in a reasonable time. Furthermore, the method supports multi-material problems and seamlessly integrates both Dirichlet and Neumann boundary conditions. In principle, the method can also be expanded into other fields of simulation on regular grids, in particular those where at least a subset of the required components can be pre-computed.

## Supporting information

**S1 Data.**
(ZIP)

## Acknowledgments

The authors would like to thank Dr. Thorsten Tjardes from the Witten-Herdecke University for providing the clinical CT data used in this paper. For the materials science samples, we would like to thank the Chair of Functional Materials at Saarland University, headed by Frank Mücklich.

## Author Contributions

**Conceptualization:** Christian Schlinkmann, Michael Roland, Philipp Slusallek, Stefan Diebels, Tim Dahmen.

**Data curation:** Christian Schlinkmann, Patrick Trampert.

**Formal analysis:** Patrick Trampert.

**Funding acquisition:** Tim Dahmen.

**Methodology:** Christian Schlinkmann, Tim Dahmen.

**Software:** Christian Schlinkmann, Michael Roland, Christian Wolff, Tim Dahmen.

**Supervision:** Philipp Slusallek, Stefan Diebels.

**Visualization:** Tim Dahmen.

**Writing – original draft:** Christian Schlinkmann, Tim Dahmen.

**Writing – review & editing:** Christian Schlinkmann, Christian Wolff, Tim Dahmen.

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
