## [Decision Letter · Decision Letter 0]

14 May 2020

PONE-D-20-11213

A GPU-Based Caching Strategy for Multi-Material Linear Elastic FEM on Regular Grids

PLOS ONE

Dear Dr Dahmen,

Thank you for submitting your manuscript to PLOS ONE. After careful consideration, we feel that it has merit but does not fully meet PLOS ONE’s publication criteria as it currently stands. Therefore, we invite you to submit a revised version of the manuscript that addresses the points raised during the review process.

The major comment (among many) is the lack of rigorous validation of the method presented as well as the observation that performance comparisons seem not to be inconsistent. Detailed reviewers' comments are attached below as well as in the pdf attachment.

We would appreciate receiving your revised manuscript by Jun 28 2020 11:59PM. To enhance the reproducibility of your results, we recommend that if applicable you deposit your laboratory protocols in protocols.io, where a protocol can be assigned its own identifier (DOI) such that it can be cited independently in the future. For instructions see: http://journals.plos.org/plosone/s/submission-guidelines#loc-laboratory-protocols

We look forward to receiving your revised manuscript.

Kind regards,

Anotida Madzvamuse, Dphil, MSc, MSC-ED, BSc (Hon)

Academic Editor

PLOS ONE

Journal Requirements:

2. Please note that PLOS ONE has specific guidelines on software sharing (http://journals.plos.org/plosone/s/materials-and-software-sharing#loc-sharing-software) for manuscripts whose main purpose is the description of a new software or software package. In this case, new software must conform to the Open Source Definition (https://opensource.org/docs/osd) and be deposited in an open software archive. Please see http://journals.plos.org/plosone/s/materials-and-software-sharing#loc-depositing-software for more information on depositing your software.

3. Please ensure that all datasets used for testing are referenced and linked to in the manuscript as well as in the Data availability statement.

4. Please ensure that you refer to Figure 5 in your text as, if accepted, production will need this reference to link the reader to the figure.

Additional Editor Comments (if provided):

Reviewers' comments:

Reviewer's Responses to Questions

**Comments to the Author**

1. Is the manuscript technically sound, and do the data support the conclusions?

Reviewer #1: Partly

Reviewer #2: Yes

Reviewer #3: Yes

2. Has the statistical analysis been performed appropriately and rigorously? 

Reviewer #1: N/A

Reviewer #2: N/A

Reviewer #3: N/A

3. Have the authors made all data underlying the findings in their manuscript fully available?

Reviewer #1: Yes

Reviewer #2: Yes

Reviewer #3: Yes

4. Is the manuscript presented in an intelligible fashion and written in standard English?

Reviewer #1: Yes

Reviewer #2: Yes

Reviewer #3: Yes

5. Review Comments to the Author

Reviewer #1: The review comments are attached as a PDF file.

Reviewer #2: The contribution is well-written, and describes an interesting application of GPU accelerated finite-element solution procedure for linear elasticity. Despite the application on the tibia, the link to the biomedical practice is very weak. I think that the manuscript should at least provide the PDE from a balance of momentum, Hooke's Law and the link to the displacements, as well as the boundary conditions. Furthermore, I think that the authors should show some computed results over the domain of computation in order to illustrate a stronger link to the biomedical practice. Next to these comments, which should not be too difficult to incorporate into this manuscript, I have the following more detailed points:

1. line 35: I would also mention 'symmetric' because the linear operator in the PDE for the momentum balance is self-adjoint,

Furthermore, only if all boundaries that are considered are free (or are force-boundary conditions), then the resulting discretisation matrix will be positive semi-definite, else it will be positive definite (the same holds for the continuous operator).

2. Regarding the text and explanation near equation (1), it may be helpful to the PONE reader to add an illustrative figure to describe the 2D case. Probably a 3D illustration could end up a bit messy.

3. line 264-5: the residual error that has been introduced in equation (2) is not an absolute but a relative residual error.

4. Line 276: I would replace 'simulation' with 'iterative procedure'

5. Line 279: In the explanation of the averaged error, it may be good to add a link to the scaled L2-norm over the portion of the domain where the error is nonzero (f <> 0), you might as well refer to the L2 functional as an integration of the portion of the domain where the error is nonzero. In fact this is what your error is. (error = 1/N sum_j (f_j <> 0) (f_j - Au_j)^2, where N represents the number of nodes where f_j <> 0. This is just a matter of terminology.

6. I would replace several occurrences of 'compute time' with 'computation time'

7. Figure 5: An absolute error in terms of a percentage is reported. How can an absolute error be expressed in terms of percentages? Representations in percentages are always relative, I think.

8. Figure 5 and the text describing the figure: Does the level for the convergence criterion depend on the highest mesh resolution used in the simulation? Please, comment on this issue. Because now the results are presented as some 'absolute numbers', which are very much depending on the problem and its discretisation.

9. The test with a randomised choice of stiffness values throughout the domain of computation are appreciated. How does the method perform if the ratio of elasticity ratios becomes very large, say 10^7? For this case, you might have to use deflation preconditioned methods. Please, comment a bit on this issue (I don't expect that the authors solve this problem, but I expect some comments and/or an example computation only).

Reviewer #3: The paper presents a matrix-free FEM simulation of linear elastic problems for applications that rely on a very high grid resolution obtained from different imaging technologies. The proposed scheme combines a representative range of pre-computed element stiffness matrices and the computing capacities of GPUs to solve the displacement equation. The presented research is very interesting and offers great potential in different computational fields. The manuscript deserves to be published after a number of improvements are taken.

As general comment, the Materials and Methods section would benefit from a more systematic and rigorous presentation of the different elements that compose it.

Additional comments, some in relation with the previous one:

1. The given description of the partition scheme (lines 187-191) may be confused with the element connectivity. It is also not sufficiently clear why the update region consists of 6x6x6 vertices.

2. Figure 2 seems to be incomplete. Consider editing or writing the pseudocode in algorithmic form.

3. The multigrid scheme should be presented in more detail, including number of refinement levels, cycle scheme, etc. What is the specific pre-defined minimum problem size mentioned in line 212? Is it the same for the three test cases given in the results? Shall the "additional levels of detail" be understood as refinement levels?

4. For the sake of completeness and fair comparison, the characteristics of the linear solver used in the Abaqus solution should be given (iterative method, tolerances, convergence criterion, ...).

5. When comparing the execution time against that using Abaqus (Table 2), is the "reduced" calculation of the residual error (on one node per cell) used? Have the execution times been measured using the full computation of the residual error? If so, why such a reduction in the computational cost should be disregarded in discussion of the above-mentioned table?

6. A detailed look at Figure 5 (the comparison of the AlSi system with the solution given by the software Abaqus) raises a number of issues that should be addressed with more detail in the text. First and most important, it seems that there is a critical value for the tolerance above which the scheme does not capture realistic features of the solution and below which the numerical solution does not improve. Is the same trend observed when the solution to the dual steel phase is compared against the Abaqus solution? A better way to address this issue might be to consider a relevant analytic test problem and perform a convergence study on it.

6. PLOS authors have the option to publish the peer review history of their article (what does this mean?). If published, this will include your full peer review and any attached files.

Reviewer #1: No

Reviewer #2: No

Reviewer #3: No

---

## [Author Response · Author response to Decision Letter 0]

10 Aug 2020

First of all, we would like to thank all reviewers for their detailed and very helpful feedback. We have prepared a revised version of our manuscript. In the following, we address the issues raised one by one.

To Reviewer #1

Major issues

1. Presenting a novel numerical method for a PDE problem without even stating the PDE is not a good practice, even in an Engineeristic context where linear elasticity is taken for granted. Moreover, the local matrices mentioned in Table 1 should be defined. To a cross-disciplinary audience, concepts such as “linear elasticity" or “stiffness matrix" are too general and may do not lend themselves to a unique interpretation. The presentation of the numerical method should be as self-contained as possible.

We have added a new section headed “Problem” to line 133ff. The section expands on the mathematical background of the simulation environment. We also considered adding the newly introduced symbols to Table 1, but decided against this as they are standard notation. The caption of Table 1 was therefore changed to reflect this. The caption now reads:

“Symbols used in the description of the algorithm.”

2. Page 25, line 323. It should be made clear from the very beginning of the Results/Performance section that, while the experiments here are carried out in single precision, Abaqus uses double precision. In this regard, Table 1 serves only to show the asymptotic behaviour of the speedup factor. By the way, does the speedup factor retain the same trend in double precision?

We thank the reviewer for this remark, which led to a number of additional experiments with surprising results. We added a subsection headed “Numerical Precision” to the results section, line 438ff. The section reads:

“In order to investigate questions of numerical precision, we repeated the experiments on all three data sets (AlSi, Dual Phase, Tibia) in both single and double precision. For the used convergence criterion, we found no measurable difference in the precision of the result. Performance was observed to be less than 10% slower when using double precision.”

We also added two paragraphs to the discussion section from line 481ff. elaborating on the interpretation of these results. The paragraphs read:

“Experiments on using different numeric precisions (single vs. double precision floating point) showed no impact on the simulation results. This shows that for our specific experiments, the simulation precision is limited by the used termination criterion, not numeric precision. It also hints at a favourable numeric stability of our method, i.e. a relatively low numerical precision is required for a system matrix of given condition number. 

Computation times were about 10% slower when using double precision. This slowdown originates entirely from the increased memory consumption, as the used P100 GPU is designed for double precision arithmetic and computation operations do not benefit from reducing floating point numbers to single precision. We speculate that the results would look differently on consumer hardware and that using double precision would have a much larger performance impact on these systems.”

Minor issues

1. Page 8, lines 26-27. “The mesh cells can have arbitrary connectivity in order to express a wide range of shapes". Not really arbitrary connectivity: in classical FEMS, the intersection of two elements cannot be a portion of a face, because hanging nodes are not allowed, as correctly explained later in the manuscript. The term “arbitrary" should be relaxed, perhaps by invoking the notion of face matching.

We agree that the use of the term ‘arbitrary’ is in this case misleading and have re-written line 27f., which now reads: 

“The mesh cells are flexible in their connectivity in order to express a wide range of shapes.”

2. Page 9. Some bibliographical references on multigrid solvers should be added at the end of the first paragraph.

We added three additional references on multigrid solvers to line 103ff.

3. Page 10, lines 72-79. Is this the first example of a matrix-free FEM for linear elasticity problems, or is this an improvement of existing studies on the topic? This is well explained later in the State-of-the-Art section, but a flavour of the exact novelty of the work should be given as soon as possible.

We added clarification of the novel contribution to line 72ff. The paragraph now reads:

“We contribute to the class of matrix-free FEM for linear elasticity problems by presenting a novel, GPU friendly caching scheme for linear elastic fixed-grid problems with a limited number of discrete materials.”

4. Page 12, line 127. “...the updated displacements of their 27 direct neighbors". The shape of elements (tetrahedral or hexahedral) being considered and even the number of space dimensions should be mentioned. Otherwise, the reader might struggle to understand why the direct neighbours are exactly 27. This is made explicit only at a later stage. Moreover, are nodes counted as “direct neighbours" of themselves?

We agree that the number 27 might be confusing to the reader at this stage. We also noticed that the exact number of neighbors is not (yet) relevant to understand the text. We therefore solved the problem by deleting the number. Line 125ff. now reads:

“By shifting the computational focus away from elements and onto per-vertex equations, vertices are iteratively updated by considering the updated displacements of their direct neighbors.”

5. Page 12, lines 129-130. “...enhancing cache efficiency by several orders of magnitude". Cache efficiency is actually discussed in the numerical experiments, but a direct comparison with the method in [34] showing the “improvement by several orders of magnitude" appears to be missing.

A direct comparison to the implementation of [38] was not possible because the authors did not release their source code or sample datasets. Our claim is therefore based on our own implementation, which implements the same Gauss-Seidel relaxation algorithm as [38], if the novel caching scheme is disabled. We added a remark to line 350 to clarify this.

While making this change, it also occurred to us that labeling the approach in [38] as “naïve” might be perceived as unfriendly. We therefor replaced the term “naïve” with “caching disabled” throughout the text.

6. Page 12, lines 133-134. “In most problems with a limited number of discrete materials". Please provide bibliographic reference(s) to such problems.

We added three references to new line 150 as requested. 

7. Page 13, line 142. “Two distinct materials are represented with gray and blue". The only visible colors in Fig. 1a) are red and blue.

We replaced “gray and blue” with “red and blue” in the caption of Figure 1.

8. Page 15, equation 1. This equation should be referenced in the text. Moreover, the whole paragraph above equation (1) (starting from “the mass matrix M") is hard to keep in the short-term memory. Hence, this discussion should be presented as a comment below equation (1), not as a preamble.

We moved several lines from the paragraph above Equation 1 that describe Equation 1 into a caption below. Three lines that are more relevant to the implementation of Equation 1 in practice were kept in the preamble, which now reads:

“If the center vertex is part of the problem boundary, then some neighboring vertices may not exist. In this case, the missing vertices are substituted with a null stiffness matrix. Crucially, in the case of linear elasticity, the matrices and boundary conditions for this vertex remain constant for all subsequent relaxations.”

9. Page 15, lines 175-176. “Neumann boundaries are stored separately as a three-component vector". Why? What information does this vector exactly contain? Is there an instance of such a vector for each vertex belonging to a Neumann boundary? Or just one vector for each Neumann face?

We agree that the formulation could be confusing to the reader, particularly the use of the term “separately”. The aforementioned line was rewritten to place more emphasis on the fact that this three-component vector is stored alongside the 27 matrices for each unique local problem configuration; it now reads (line 186ff):

“Each unique local problem configuration also contains a three-component vector describing the external force incident at the center vertex, corresponding to any Neumann boundaries that were assigned to this vertex.”

10. Page 16, lines 202-205. The difference between the discrete problem and the vertex-based representation should be made more clear.

We expanded the description of the discrete and vertex-based representations to more clearly define their unique roles in the solver. Lines 217ff. now read: 

“This representation is called the discrete problem, which holds the material assignments of all hexahedral elements as well as the boundary conditions of the problem. The discrete problem is an intermediate representation not used in the iterative solver, instead serving to assist in the pre-processing of the multigrid structure and the local problem configurations. A vertex-based representation is generated from the nodes of the discrete problem in which vertices, rather than hexahedral elements, are stored.”

11. Page 17, Figure 2. The pseudocode text is cut. To avoid this, the pseudocode should be inserted as text instead that as a figure.

We have reformatted the pseudocode text to ensure that it is readable.

12. Page 29, lines 443-445. “While the experiments on the increasing numbers of materials are somewhat artificial, they serve to show that the caching strategy continues to be beneficial in cases where the original assumption of an ordered distribution of materials holds". maybe the authors mean “...an ordered distribution of materials does not hold anymore"?

We agree that the formulation was not sufficiently clear in distinguishing the ordered and unordered distribution cases. We have rewritten the lines (494ff.), which now read:

“While the experiments on the increasing numbers of materials are somewhat artificial, they serve to show that the caching strategy continues to be beneficial in cases where the number of materials increases as long as the assumption of an orderly distribution of materials still holds. However, a random distribution of a high number of unique materials defeats the caching strategy and nullifies any performance benefits.”

To Reviewer #2

1. line 35: I would also mention 'symmetric' because the linear operator in the PDE for the momentum balance is self-adjoint. Furthermore, only if all boundaries that are considered are free (or are force-boundary conditions), then the resulting discretisation matrix will be positive semi-definite, else it will be positive definite (the same holds for the continuous operator).

We have added the terms “symmetric” and “positive definite” to the description of the stiffness matrix as suggested. The line now reads:

“Typically it is symmetric, positive definite and often very sparse, as each finite element is typically incident to only a small number of other elements.”

2. Regarding the text and explanation near equation (1), it may be helpful to the PONE reader to add an illustrative figure to describe the 2D case. Probably a 3D illustration could end up a bit messy.

We have added a cross-reference to Figure 1 (c) which shows a 3D illustration of the local problem described by Equation 1. 

3. line 264-5: the residual error that has been introduced in equation (2) is not an absolute but a relative residual error.

This is indeed a relative error. We have replaced the term ‘absolute’ with ‘relative’ as suggested.

4. Line 276: I would replace 'simulation' with 'iterative procedure'

The line was changed as suggested.

5. Line 279: In the explanation of the averaged error, it may be good to add a link to the scaled L2-norm over the portion of the domain where the error is nonzero (f <> 0), you might as well refer to the L2 functional as an integration of the portion of the domain where the error is nonzero. In fact this is what your error is. (error = 1/N sum_j (f_j <> 0) (f_j - Au_j)^2, where N represents the number of nodes where f_j <> 0. This is just a matter of terminology.

We agree that the suggested notation is more precise but found that the new notation is too complex for an in-line presentation. We therefore extracted a separate formula to line 294ff.

6. I would replace several occurrences of 'compute time' with 'computation time'

All occurrences were replaced as suggested.

7. Figure 5: An absolute error in terms of a percentage is reported. How can an absolute error be expressed in terms of percentages? Representations in percentages are always relative, I think.

We have replaced the term ‘absolute’ with ‘relative’ in both Figure 5 and its caption.

8. Figure 5 and the text describing the figure: Does the level for the convergence criterion depend on the highest mesh resolution used in the simulation? Please, comment on this issue. Because now the results are presented as some 'absolute numbers', which are very much depending on the problem and its discretisation.

We expanded the caption of Figure 5 to more clearly describe how the error values for the convergence plots are calculated. The line in question now reads:

“a) A solution for the AlSi dataset was computed with Abaqus and considered as ground-truth for this purpose. The mean relative error of the displacements as compared to this ground-truth solution was calculated on the full resolution detail level and plotted as a function of the convergence criterion.”

9. The test with a randomised choice of stiffness values throughout the domain of computation are appreciated. How does the method perform if the ratio of elasticity ratios becomes very large, say 10^7? For this case, you might have to use deflation preconditioned methods. Please, comment a bit on this issue (I don't expect that the authors solve this problem, but I expect some comments and/or an example computation only).

We have expanded the relevant section with the following comments on this problem:

“In addition, tests of materials with very large differences in elasticity ratios, for example on the order of 107, have shown similar computational errors as in other FEM approaches. For such cases a deflation preconditioner may need to be used.”

To Reviewer #3

1. The given description of the partition scheme (lines 187-191) may be confused with the element connectivity. It is also not sufficiently clear why the update region consists of 6x6x6 vertices.

We agree that the size of the update region is not clearly described at this stage. Upon review we feel that this information is an implementation detail that is not relevant at this stage in the text, and is already described more thoroughly in the Implementation section. As a result, we decided to remove this line. We also added a line (203ff.) to emphasize the difference between the partitioning scheme and the element connectivity:

“This partitioning refers to the order of computation and should not be confused with element connectivity.”

2. Figure 2 seems to be incomplete. Consider editing or writing the pseudocode in algorithmic form.

We have adjusted the formatting of the pseudocode text to ensure that it is readable.

3. The multigrid scheme should be presented in more detail, including number of refinement levels, cycle scheme, etc. What is the specific pre-defined minimum problem size mentioned in line 212? Is it the same for the three test cases given in the results? Shall the "additional levels of detail" be understood as refinement levels?

We have expanded the description of the multigrid scheme used in our solver. The relevant section now reads:

“These steps are repeated for each level of the multigrid, down to a minimum problem size encompassing the size of one update region. We employ a half V-Cycle multigrid as the simplest possible approach that is compatible with our solver. Each level of the multigrid is solved completely before projecting the resulting displacements down to the next level. Once a coarse level has converged and the result projected to the next finer level of detail, the associated data may be discarded. In principle, more elaborate multigrid cycle schemes such as F-Cycles or W-Cycles are probably also compatible with our solver.”

4. For the sake of completeness and fair comparison, the characteristics of the linear solver used in the Abaqus solution should be given (iterative method, tolerances, convergence criterion, ...).

We have added additional information on the characteristics of the linear solver used in the Abaqus simulations to the Evaluation Environment section, which now reads as follows:

“The simulation was performed in parallel on a workstation with 16 CPU cores and 128 GB of RAM. The linear solver was configured to use the iterative method with a convergence criterion of 5.0×10-3 for the average flux norm and 1.0×10-2 for displacement corrections.”

5. When comparing the execution time against that using Abaqus (Table 2), is the "reduced" calculation of the residual error (on one node per cell) used? Have the execution times been measured using the full computation of the residual error? If so, why such a reduction in the computational cost should be disregarded in discussion of the above-mentioned table?

The evaluation of a termination criterion is an integral part of any iterative numerical procedure. Therefore, the computation of the termination criterion “residual error” (including all optimizations thereof) is included in the reported results in Table 2 for all methods. Computing more than the required residual error values would slow down computation with no perceivable advantage, so the possibility is not discussed.

6. A detailed look at Figure 5 (the comparison of the AlSi system with the solution given by the software Abaqus) raises a number of issues that should be addressed with more detail in the text. First and most important, it seems that there is a critical value for the tolerance above which the scheme does not capture realistic features of the solution and below which the numerical solution does not improve. Is the same trend observed when the solution to the dual steel phase is compared against the Abaqus solution? A better way to address this issue might be to consider a relevant analytic test problem and perform a convergence study on it.

We thank the reviewer for this very helpful remark. As not all of the implementation details of the commercial software Abaqus are exposed, the exact cause of the discrepancy between our solver and the reference solution remains unknown. In order to exclude a systematic bias in our solver as an explanation, we performed experiments against analytically known standard problems. A new section (line 385ff.) and Figure 5c were added to include these results. The caption of Figure 5 was extended accordingly.

---

## [Decision Letter · Decision Letter 1]

5 Oct 2020

A GPU-Based Caching Strategy for Multi-Material Linear Elastic FEM on Regular Grids

PONE-D-20-11213R1

Dear Prof. Dr. Dahmen,

We’re pleased to inform you that your manuscript has been judged scientifically suitable for publication and will be formally accepted for publication once it meets all outstanding technical requirements.

Kind regards,

Anotida Madzvamuse, Dphil, MSc, MSC-ED, BSc (Hon)

Academic Editor

PLOS ONE

---

## [Editor Report · Acceptance letter]

21 Oct 2020

PONE-D-20-11213R1 

A GPU-based caching strategy for multi-material linear elastic FEM on regular grids 

Dear Dr. Dahmen:

I'm pleased to inform you that your manuscript has been deemed suitable for publication in PLOS ONE. Congratulations! Your manuscript is now with our production department. 

Kind regards, 

on behalf of

Professor Anotida Madzvamuse 

Academic Editor

PLOS ONE